# Investigating Airline Service Quality from a Business Traveller Perspective through the Integration of the Kano Model and Importance–Satisfaction Analysis

Patricia Lippitt [1], Nadine Itani [1,*], John F. O'Connell [1], David Warnock-Smith [2] and Marina Efthymiou [3]

1   Centre for Aviation Research, University of Surrey, Guildford GU2 7XH, UK
2   School of Aviation and Security, Buckinghamshire New University, High Waycombe HP11 2JZ, UK
3   DCU Business School, Dublin City University, D09 RFK0 Dublin, Ireland
*   Correspondence: n.itani@surrey.ac.uk

**Abstract:** This study uses the Kano model and importance–satisfaction analysis (ISA) to assess airline service quality by identifying the prioritised service quality attributes (SQA) for business travellers. The study aims to produce suggestions for airline executives on how to allocate resources in the most effective way to enhance the quality of service and increase customer satisfaction. A conceptual framework divides business travellers into four Clusters based on the behavioural variables of flight length and cabin class. For each Cluster, business traveller expectations for fourteen SQAs were assessed through using the Kano model while integrating the ISA. The empirical phase employs a 38-item questionnaire that was shared on various frequent flyer and business travel forums. Additionally, this study utilises an adapted qualitative questionnaire where four airline managers expressed their perceptions on how they think business travellers perceive the fourteen SQAs. The analysis reveals four categories, namely 'concentrate here', 'keep up the good work', 'low priority', and 'possible overkill', exhibiting the importance and satisfaction of the fourteen SQAs. Findings show that resource allocation was adequate on only five attributes out of fourteen. The analysis of the airline manager responses shows differences in their assessment when compared to business travellers for two tangible attributes.

**Keywords:** business traveller; service quality; customer satisfaction; Kano model; importance satisfaction analysis (ISA)





## 1. Introduction

With the rise of competition in the airline sector, it has become increasingly crucial to prioritise offering excellent service and creating value for customers [1–3]. Value creation is linked to the relational capital that increases profitability by enhancing customer loyalty and satisfaction [4]. It is generally understood that customer satisfaction increases as perceived service quality increases [5]. It is commonly acknowledged that the connection between the various aspects of service quality and customer satisfaction might not follow a linear trend, indicating that superior service quality may not necessarily result in greater customer satisfaction [6]. Therefore, it is necessary to classify the components of service quality based on how they impact customer satisfaction [7].

Airlines are anticipated to meet passengers' expectations on all service quality attributes (SQA) to become the preferred airline [8,9]. However, research indicates that there is still a lack of complete comprehension regarding customer expectations and perceptions of airline services [10,11]. Furthermore, numerous airline operators struggle with proper resource allocation [12].

Understanding customer expectations and needs is a critical factor to ensure service sustainability in the airline industry [13,14]. Airline operators must give precedence to service traits that significantly influence essential customer expectations and requirements,

and effectively manage their resources to adequately fulfil those expectations [5,10]. Additionally, customer expectations regarding service quality differ between consumer segments. An important segment for airlines is business travellers which accounts for high amounts of revenues with worldwide business travel expenditure reaching approximately USD 1300 billion in 2019 [15,16]. Nonetheless, there are relatively few studies in the airline industry that assess the comparative significance of service quality aspects on the satisfaction levels of business travellers [17,18].

Therefore, this research aims to critically investigate airline service quality from a business traveller perspective to identify the highest priorities for airline business passengers and accordingly optimise airline resource allocations through clustering and integrating the Kano model with importance–satisfaction analysis (ISA). The conceptual framework applied in this research is partially based on a novel approach in Tahanisaz and Shokuhyar [19]. There are very few studies investigating the importance of airline SQAs exclusively from a business traveller perspective. This study adds to the limited existing research providing airline managers with information on business passengers' expectations, their level of fulfilment, and the impact of these expectations on airline operator resource capacity and allocation.

Additionally, the study aims to answer the research question: What are the most critical SQAs for business travellers? The following research objectives were identified: (1) Investigate existing research on airline service quality for business travellers; (2) assess a sample of empirical data on business traveller service quality by utilising the Kano model and ISA; (3) establish behavioural differences regarding SQAs for the business travel market segment; and (4) determine if airline managers are correctly resourced and resonating with the identified requirements of business travellers.

The structure of this study is as follows: Section 2 provides the review of the literature; Section 3 describes the methodology, the conceptual framework, and the design of the research instrument; the results and discussion are provided in Section 4, while Section 5 focuses on the conclusions and the managerial implications.

## 2. Review of the Literature

### 2.1. Service Quality and Customer Satisfaction

Service quality has attracted considerable interest and debate because of the difficulties in both defining and measuring it with no general agreement on either [20]. Several definitions exist describing the meaning of service quality, the most common one is Parasuraman et al. [21], who define service quality as "The discrepancy between consumers' perceptions of services offered by a particular firm and their expectations about firms offering such services". Lewis and Mitchell [22] and Dotchin and Oakland [23] define service quality as being the extent to which a service meets customer needs or expectations. According to Woodside et al. [24], service quality is regarded as customer evaluations of the services provided by organisations. Park et al. [25] define service quality as a consumer's overall impression of the efficiency of an organisation and its services or the chain of services divided into a series of processes [13]. It is, therefore, important for organisations to prioritise the quality of service provided to customers, since customer feedback is reflected through trust and behavioural intentions towards the products and services. Bateson and Hoffman [26] suggested that a customer's overall evaluation of the service provider's performance is realised through the cognitive formation of service quality.

Prajogo and McDermott [27] reveal that organisations differentiate themselves through service quality and gain a long-lasting competitive edge over their rivals. Therefore, organisations maintain a high level of service quality in markets where the intensity of rivalry is considerably high [28]. Most definitions of service quality depend on the context and therefore focus on how well the service delivered matches the customer's expectations.

On the other hand, customer satisfaction is mainly derived from the physiological response with the perceptual difference gap between expectation before consumption and practical experience after consumption. According to Oliver [29], satisfaction is the emotion

of contentment or discontentment that arises when a product's perceived performance is compared to expectations. On the other hand, Hansemark and Albinson [30] define customer satisfaction as a general attitude towards a service provider, or an emotional response to the gap between what customers expect and what they actually receive in terms of fulfilling their needs. The existing customer satisfaction literature is often dominated by SERVQUAL [21], which recognises the significance of service quality as a driver of customer satisfaction. According to Caruana [31], service quality positively impacts customer satisfaction; therefore, quality service delivered by the organisation makes a customer feel satisfied.

The discrepancy between expected and experienced service attributes has also been quite extensively reviewed within the context of public transport. Gao, Rasouli, Timmermans, and Wang [32], for instance, used public transport satisfaction data from Xian in China to conclude that the perceived 'service gap' explanation of customer satisfaction is also influenced by the attitudes, moods, and personality elements of travellers themselves. These findings were also extended by Sarker et al. [33] who tested public transport user reactions to service disruption.

### 2.2. Airline Service Quality, Customer Satisfaction, and Loyalty

While several studies discuss the measurement of airline service quality, there is no consensus on what criteria are most relevant to attract and satisfy airline customers, their preference for an airline operator or one flight over another, or to assess service quality [9,34–36]. Eboli et al. [37] in their review of the literature suggest that most studies investigate airlines' service attributes in three different phases: before, during, and after the flight. Commonly, the service attributes are divided into two main service dimensions: empathy (including attributes regarding how the company cares for and provides individualised attention to their customers), and tangibles (including cleanliness of aeroplane interior and toilets, quality of catering, and comfort level of the plane seats). A preferred method for determining airline SQAs is adopting the "SERVQUAL scale" developed by Parasuraman et al. [38], which is a widely employed instrument to measure service quality. Reliability, assurance, tangibles, empathy, and responsiveness were emphasised as five important dimensions of airline service quality in various studies [39–42].

Previous studies have suggested additional criteria for determining the attractiveness of airlines beyond the main dimensions. Medina-Muñoz et al. [9] indicated that 'safety and punctuality', 'ticket price', and 'attention and service during the journey' were the most important criteria in determining airline attractiveness. Kim and Park [43] emphasised the importance of professional knowledge of cabin crew, emergency handling, and flight schedules for airline service quality. Pangow et al. [44], Munusamy et al. [45], and Mikulić and Prebežac [46] stressed the relevance of cabin crew service quality. Kurtulmuşoğlu et al. [47] and Chen and Chao [34] highlighted the significance of punctuality when evaluating airline service quality.

Surovitskikh and Lubbe [48] emphasised the significant effect of on-time performance (OTP) on the consistency of service quality. Gilbert and Wong [40] detailed dimensions of reliability, assurance, facilities, employees, flight patterns, customisation, and responsiveness as the core components of airline service quality. Kim et al. [49] put forward an airline's safety record, ticket prices, cabin food/beverages, and average delay times as effective service quality criteria for airline selection. Elliott and Roach [50] used OTP, baggage handling, food quality, seat comfort, check-in services, and in-flight services to define airline service quality.

The relationship between airline service quality and passenger satisfaction has been researched extensively. Studies by Ali et al. [51] and Farooq et al. [52] showed that the service quality dimensions (SQDs) of airline tangibles, terminal tangibles, personnel services, empathy, and brand image have a direct, positive, and strong effect on passenger satisfaction.

The analysis by Carvalho and Medeiros [53] on airlines in Brazil revealed that reliability, responsiveness, and assurance did not show a considerable effect on customer satisfaction. This contrasts with the findings by Akram et al. [54]. This overview of studies focusing on the relationship between SQDs/SQAs and customer satisfaction in the airline industry indicates that airline service quality is closely related to passenger satisfaction and is often regarded as the foundation of customer satisfaction [55,56].

Increasing amounts of studies have assessed the relationship between airline service quality, passenger satisfaction, and loyalty on behavioural intentions [46,57,58]. Research by Namukasa [59], for instance, indicated that service quality is a considerable driver for passenger satisfaction and, in turn, has a considerable impact on customer loyalty.

## 3. Research Methodology

### 3.1. Conceptual Framework

This study applies segment clustering, the Kano model, and importance–performance analysis (IPA) to assess airline service quality by identifying the most important priorities of SQAs for business travellers.

Clustering: airline passengers are clustered based on behavioural variables to identify distinctive groups with similar expectations. Tahanisaz and Shokuhyar [19] mention that previous studies utilised the Recency, Frequency, and Monetary model to cluster customers, whereas they proposed the Flight Intent, Cabin Class, and Frequency model. This study clusters business travellers utilising the following variables: (1) flight length: short-haul or long-haul, and (2) cabin class: economy or business class. Hence, business travellers are categorised into four clusters as shown in Figure 1.

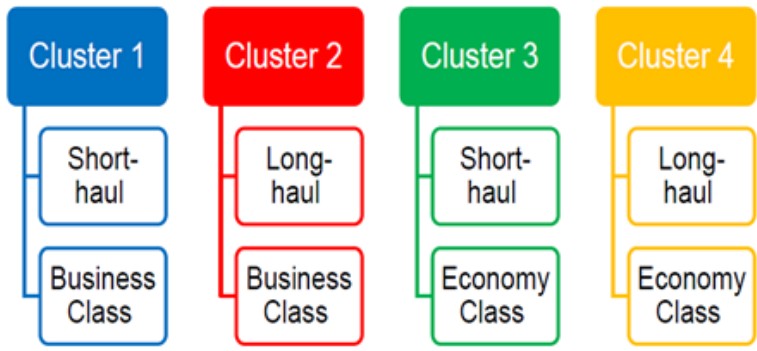

**Figure 1.** Business traveller clusters.

Kano model: The Kano model was originally used to improve the development of product quality in manufacturing, but it has since been applied in various service industries [60]. The Kano model and SERVQUAL are commonly utilised in the service sector to attain customer satisfaction in healthcare, tourism, banking, and education [61–65].

Although the use of the Kano model in exploring the underlying factors that impact airline service quality is not very common, some studies in the airline service quality literature have utilised it [7]. Liou et al. [5] demonstrated in a Taiwanese airline case study how the Kano model can be used to analyse airline passenger service requirements and discussed the potential benefits of using this approach to establish a marketing strategic plan. Shahin and Zairi [66] also applied the Kano model to customer requirements in the airline service industry and suggested that airlines should use it to identify the most critical customer requirements and cope with the dynamic nature of the highly competitive service market environment.

Airlines face the conundrum of having to enhance services whilst seeking to keep costs low and operating on slim margins [67]. Through the implementation of the Kano model, management decisions on investments to improve service quality can be enhanced and resource allocation optimised [66]. Figure 2 illustrates the Kano model, with the x-axis

representing the number of quality aspects and the y-axis illustrating the satisfaction of business travellers. The model divides SQAs into five categories, described below [19].

- Cat. 1. Must-be attributes are fundamental features for business travellers. Subsequently, when an airline fails to fulfil must-be requirements, passengers will be strongly dissatisfied, whereas the fulfilment of the attribute will not cause an increase in satisfaction.

- Cat. 2. When an airline delivers one-dimensional attributes, business traveller satisfaction will be created. When these requirements are insufficient or not fulfilled, however, dissatisfaction will be caused.

- Cat. 3. Attractive attributes will produce satisfaction when delivered, yet will not cause dissatisfaction when insufficiently or not fulfilled. These attributes can be strategically significant in assisting airlines to achieve a competitive advantage over their competition.

- Cat. 4. Indifferent attributes do not significantly contribute towards satisfaction, whether they are existent or absent in the airline product, so could theoretically be removed.

- Cat. 5. Finally, reverse attributes cause dissatisfaction when fulfilled, yet lead to satisfaction when not fulfilled. Hence, an attribute in this classification should be removed.

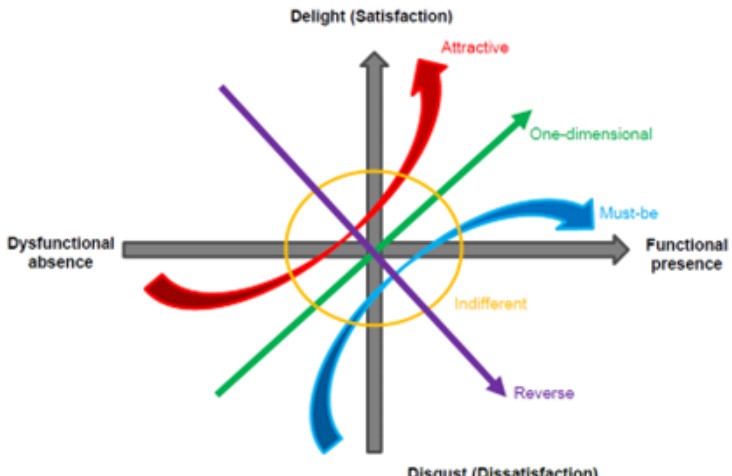

**Figure 2.** The Kano model [6].

Importance–performance analysis (IPA): Although the Kano model classifies SQAs, it does not measure their quantitative or qualitative performance [68]. Huang [69] conducted a study to examine how air passengers make decisions, using a conceptual model that took into account factors such as service value, airline service quality, satisfaction, perceived sacrifice, and behavioural intentions. The study employed structural equation modelling (SEM) and importance–performance analysis (IPA) to analyse the data [52,70].

Martilla and James [71] introduced IPA as a technique through which a company can achieve customer satisfaction through the quality attributes of products and services. In the IPA results, it was indicated that responsiveness is the most important airline SQA, and other attributes, such as complaints handling are nearly always classified as must-be, making it impossible to prioritise them [66]. Accordingly, this research integrates the Kano model into an IPA to improve attribute prioritisation. Figure 3 reveals the IPA model by Martilla and James [71], which determines the importance of SQAs and demonstrates the degree of satisfaction simultaneously. It plots results in graphic form on a matrix with two dimensions. The vertical axis shows the importance of a specific attribute, whereas the horizontal axis demonstrates the level of satisfaction. The matrix consists of four quadrants, explained below.

- Quadrant I: Customers perceive these attributes as very important. However, the performance of airlines in these attributes is regarded as below average, implying that efforts to improve should be concentrated here [13].
- Quadrant II: Attributes in this quadrant are important and customers rate their performance highly; therefore, airlines should keep up the good work [72].
- Quadrant III: Contains low-performance attributes, yet they are not very important to business travellers; therefore, these features are a low priority for airlines and limited resources should be allocated to them [4,13].
- Quadrant IV: This quadrant contains attributes that have a comparatively high performance, but the attributes are of low importance to business travellers. Airline effort on these attributes is over-utilised, suggesting possible overkill; hence, airlines should consider reallocating resources somewhere else [4,13].

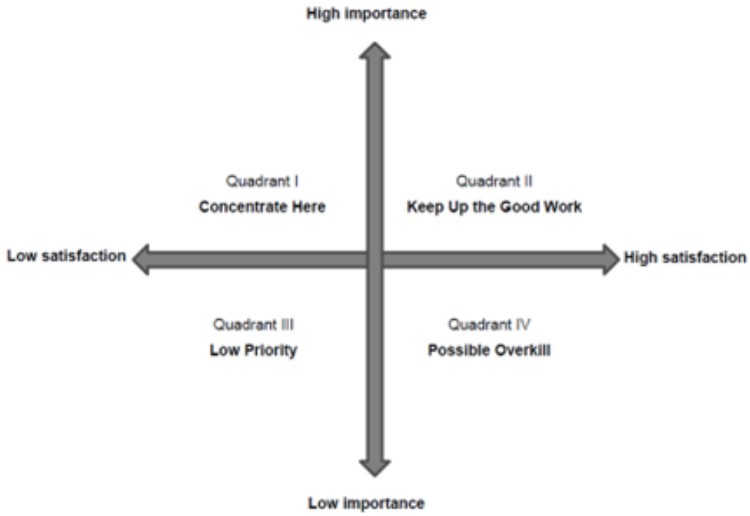

**Figure 3.** Importance–performance analysis (IPA) matrix [71].

### 3.2. Data Collection Method

Data are collected using both primary and secondary data sources. Primary data are collected using a self-administered electronic questionnaire consisting of 38 questions in three sections. The first section collects demographic and behavioural information to cluster passengers, while sections two and three gather information on the opinions of business travellers on the importance and performance levels of various SQAs. By using the questionnaire, attributes are classified into Kano categories and the data collected are used to measure how passengers perceive the importance and their satisfaction with various SQAs to perform the IPA.

For reliability and validity, the primary results are cross-checked in the discussion with external secondary literature sources, which are both subscription-based and open-sourced on the internet and from broadsheet newspapers and industry publications (*Business Travel Magazine*, Skift reports).

### 3.3. Design of the Instrument

The population is made up of passengers who previously travelled on commercial aeroplanes for work or business purposes. The target segment is business travellers worldwide, with no geographic limitations. The authors aimed to accomplish a minimum sample size of 250. A total of 339 responses were obtained and 265 of these were acknowledged as usable and were therefore utilised. Hence, the sample size of 265 was considered adequate and valid. This study uses convenience sampling that is taken from a segment of the population that is accessible. Even though the data collected from a convenience sample does not allow definite findings and the research cannot be generalised [73], convenience sampling

utilising the Internet enables the researchers to reach the difficult-to-locate population of business travellers. Additionally, to validate the questionnaire, pilot testing was conducted utilising a small sample of 25 frequent travellers. Modifications were made to the final questionnaire to prevent ambiguities and confusions after running the pilot test. Usable responses from four airline managers are collected.

The questionnaire was posted online from 29 June to 03 August 2021 in the *Business Traveller Magazine* forum, one of the leading publications for business travellers worldwide. It was also posted on frequent flyer forums, including FlyerTalk and InsideFlyer, the Business Travel section of the Reddit forum, and the aviation forums Airliners.net and Aviation24.be during the same period. Additionally, it was shared on the LinkedIn group Flying Business Travellers and the Facebook group Business Class & First-Class Flight Deals. An adapted questionnaire was created to ask airline managers to express their perceptions on how they think business travellers feel about the same SQAs. Both questionnaires used closed-ended questions. The link to the adapted self-administered questionnaire was sent via e-mail to five airline managers with four replies collected. Qualtrics software was used for questionnaire design and data analysis. From the airline service quality review of the literature, fourteen quality attributes were identified and included in the questionnaire, as displayed in Table 1.

**Table 1.** Service quality attributes.

|  | Service Quality Attributes (SQAs) | Service Quality Dimensions (SQDs) |
| --- | --- | --- |
| SQA 1 | Convenient flight schedules and frequencies | Empathy |
| SQA 2 | On-time performance (OTP) | Empathy |
| SQA 3 | Frequent-flyer programme | Empathy |
| SQA 4 | Airline responsiveness to complaints | Assurance |
| SQA 5 | Airline lounge service | Reliability |
| SQA 6 | Seat comfort | Tangible |
| SQA 7 | Cleanliness of the aircraft interior and seats | Tangible |
| SQA 8 | Appearance of cabin crew | Tangible |
| SQA 9 | Politeness of cabin crew | Tangible |
| SQA 10 | Knowledge and experience of cabin crew | Assurance |
| SQA 11 | Variety and quality of in-flight meals and drinks | Assurance |
| SQA 12 | Variety and quality of in-flight entertainment | Tangible |
| SQA 13 | Access to in-flight Wi-Fi | Tangible |
| SQA 14 | Access to in-seat power | Tangible |

The second part consists of the Kano scale. In this section, for every SQA, business traveller responses are assessed by both a functional and dysfunctional question, as shown in Figure 4, to classify attributes into six categories: Must-be (M), One-dimensional (O), Attractive (A), Indifferent (I), Reverse (R), or Questionable (Q).

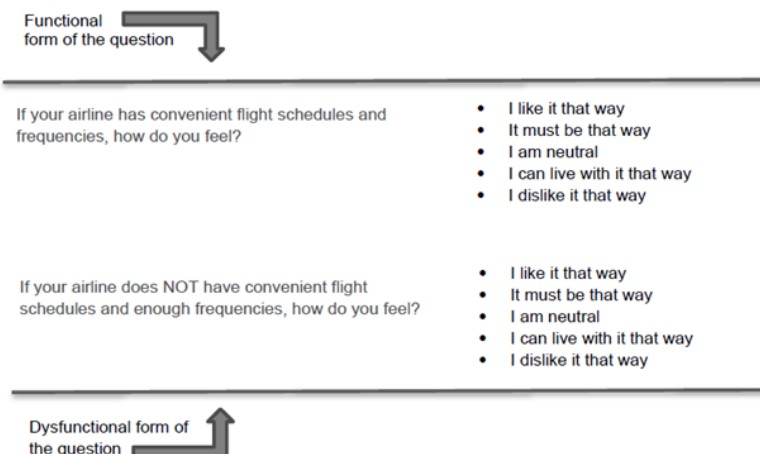

**Figure 4.** Functional and dysfunctional question examples in the Kano questionnaire (adapted) [74].

The Kano evaluation matrix shows how the SQAs are categorised by combining the two responses, as shown in Figure 5.

| Product requirement | | Dysfunctional form of the question | | | | |
|---|---|---|---|---|---|---|
| | | 1. I like it that way | 2. It must be that way | 3. I am neutral | 4. I can live with it that way | 5. I dislike it that way |
| Functional form of the question | 1. I like it that way | Q | A | A | A | O |
| | 2. It must be that way | R | I | I | I | M |
| | 3. I am neutral | R | I | I | I | M |
| | 4. I can live with it that way | R | I | I | I | M |
| | 5. I dislike it that way | R | R | R | R | Q |

**Figure 5.** Kano evaluation matrix.

The final section contains the IPA questions, asking for the business travellers' evaluation regarding the performance and importance of each attribute. Likert scales are frequently used in questionnaires; normally these scales have five or seven different points. Therefore, participants are required to respond to questions by choosing between five different points to rate the performance of their airline and to express the level of importance.

To allocate each SQA into one of the Kano categories for each Cluster, the mode of the number of respondents is utilised [7]. Matzler and Hinterhuber [74] evolved the Kano model by developing a customer satisfaction coefficient. Their assessment indicates "how strongly a product/service feature may influence satisfaction or, in the case of its non-fulfilment, customer dissatisfaction". The Kano model is quantified by Wang [75] by presenting a set of three formulas: $(d_j^+)$ illustrates the functional presence (positive, delight) of an SQA; $(d_j^-)$ signifies the dysfunctional absence (negative, disgust); and $(S_j)$ describes the rate of SQA satisfaction.

$$d_j^+ = \frac{A_j + O_j - R_j}{A_j + O_j + M_j + R_j + I_j} \tag{1}$$

$$d_j^- = -\frac{M_j + O_j - R_j}{A_j + O_j + M_j + R_j + I_j} \tag{2}$$

$$S_j = P_j d_j^+ + (1 - P_j) d_j^- \tag{3}$$

$A_j, O_j, M_j, R_j$, and $I_j$ signify the percentage of the Kano categories for SQA $j$ [72]. To establish the importance and performance of each SQA, ref. [60] suggests the 'mean' to be used. Thereafter, the performance rating ($P_j$) of SQA $j$ is standardised to a value between 0 and 1. Figure 6 reveals that the degree of satisfaction ($S_j$) is determined by interpolating the two endpoints ($d_j^+$ and $d_j^-$). Consumer satisfaction is thus a weighted average of positive delight ($d_j^+$) and negative disgust ($d_j^-$).

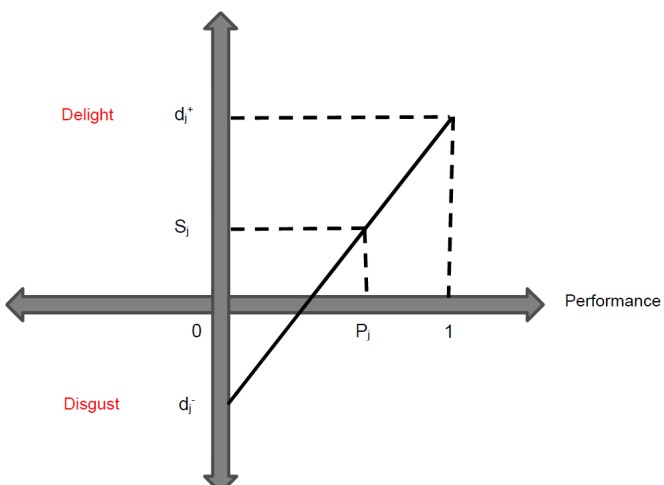

**Figure 6.** A plot to convert an attribute's performance into a satisfaction degree [72].

Thus, the participants' performance rating is converted through the Kano model, according to 'equation 3' to acquire a satisfaction degree with each attribute. To plot the IPA matrix, Wang [72] encourages the use of the median over the mean to establish the thresholds of importance and performance dimensions. Consequently, the IPA is changed into an importance–satisfaction analysis (ISA). To plot the ISA matrix, importance and satisfaction data for each Cluster are exported into SPSS, for the creation of scatterplots. The median is used to establish the thresholds of the importance and satisfaction dimensions to create an ISA matrix for each Cluster.

To ensure reliability, each participant is asked the same series of questions in a self-administered online survey [76,77]. Questions are worded carefully to make sure they meant the same to all respondents [78], and answers that represent an appropriate response are communicated consistently to all participants. Additionally, to test the reliability of the instrument scale, Cronbach's alpha is calculated in SPSS for the different sets of questions [60]. The Cronbach's alpha coefficient is higher than 0.70 for all sections, which indicates that the tested instrument scale is reliable [79].

## 4. Results and Discussion

### 4.1. Cluster Analysis

Concerning the demographic profile of the respondents, the majority are male (91.7%), between 30 and 59 years (81.2%), and are based in Europe (81.9%) with no identification of the country they live in. More than 72% of the participants fly more than five times a year,

while more than half of them (51.3%) commonly fly with British Airways. However, some respondents commonly fly short-haul (49.8%), whereas others usually fly long-haul (50.2%).

Business travellers with the same behavioural characteristics were categorised into four clusters (Figure 7). Cluster one includes travellers on short-haul business class flights. Most of them are men between 40 and 49 years old. Notably, this cluster only consists of 38 out of the total 265 questionnaire participants, suggesting that most business travellers fly in economy class on short-haul flights. This is likely because short-haul in-flight elements, including in-flight food and beverages, are less important [80]. Cluster two includes travellers flying long-haul business class. Similarly, most respondents are men between 40 and 49 years old. This cluster consists of 110 out of the 265 questionnaire participants, the highest number of respondents out of the four clusters (Figure 7). This indicates that most business travellers fly business class on long-haul trips, likely because on longer flights the in-flight comfort and the experience features become more critical.

| Clusters | Descriptions (in percentages) | | | | | | | | | Number of respondents |
|---|---|---|---|---|---|---|---|---|---|---|
| | Gender | | | Age | | | | | Cabin Class | |
| | Male | Female | Prefer not to say | 18 – 29 | 30 – 39 | 40 – 49 | 50 – 59 | 60 + | Number of respondents in:<br><br>Business Class (J)<br>Premium Economy (W)<br>Economy (Y) | |
| C1 | 97.4% | 2.6% | 0% | 15.8% | 26.3% | 31.6% | 13.2% | 13.2% | J 38 | 38 |
| C2 | 91.8% | 7.3% | 0.9% | 3.6% | 26.4% | 33.6% | 20.9% | 15.5% | J 110 | 110 |
| C3 | 88.3% | 11.7% | 0% | 8.5% | 41.5% | 25.5% | 18.1% | 6.4% | W 8 & Y 86 | 94 |
| C4 | 95.7% | 4.3% | 0% | 8.7% | 30.4% | 30.4% | 21.7% | 8.7% | W 9 & Y 14 | 23 |
| Total | | | | | | | | | | 265 |

**Figure 7.** Profile of each business traveller cluster.

Cluster three comprises business travellers who fly short-haul economy class. Unlike the previous two clusters, it has the highest number of female respondents and, noticeably, most respondents are between 30 and 39 years old. Hence, a larger proportion of female and younger business travellers fly short-haul flights in economy class. This cluster consists of 94 out of the 265 questionnaire participants. Additionally, 18 respondents (19.1%) indicated that they commonly travel with an LCC and most of them mentioned easyJet as their choice (Figure 7). This might be because easyJet has been improving the experience for business travellers on short-haul routes by introducing attributes such as Flexi rates and speedy boarding. The last cluster entails business travellers who fly in long-haul economy class. It consists mostly of men between 30 and 49 years old. Figure 7 further reveals that eight respondents (8.5%) in Cluster three and nine respondents (39.1%) in Cluster four mentioned that they travel in premium economy. This indicates that especially on long-haul flights, a significant amount of the more price-sensitive business travellers seek better comfort over the standard economy and therefore fly in the premium economy cabin.

### 4.2. Kano Model Results

For every cluster, each of the fourteen SQAs is classified into one of the five categories of the Kano model (Table 2). To allocate each SQA into one of the Kano categories, for each cluster the Mode of the number of respondents was utilised [7]. As suggested by Kuo et al. [60] the classification that would have the greatest influence on a service should be selected. Hence, the SQA should be classified in order of M > O > A; R > I. Therefore, the SQA was categorised as a must-be.

The attribute of convenient flight schedules and frequencies is a fundamental feature for business travellers in Clusters two, three, and four. Accordingly, airlines must provide convenient flight schedules at a satisfactory level for business travellers because negative performance can have a significant impact and cause dissatisfaction. This is in line with the annual surveys by IATA and OAG, which have shown that flight frequency, timings, and direct, non-stop flights are critical elements for business travellers [80].

**Table 2.** Kano categories and service quality attributes.

| Service Quality Attribute | Kano Category | | | |
|---|---|---|---|---|
| | Cluster 1 | Cluster 2 | Cluster 3 | Cluster 4 |
| SQA1. Convenient flight schedules and frequencies | O | M | M | M |
| SQA2. On-time performance | O | O | O | O |
| SQA3. Frequent-flyer programme | M | M | A | M |
| SQA4. Airline responsiveness to complaints | M | M | O | O |
| SQA5. Airline lounge service | O | M | A | I |
| SQA6. Seat comfort | O | M | O | M |
| SQA7. Cleanliness of the aircraft interior and seats | M | M | M | O |
| SQA8. Appearance of cabin crew | I | O | I | I |
| SQA9. Politeness of cabin crew | O | O | M | O |
| SQA10. Knowledge and experience of cabin crew | A | A | I | A |
| SQA11. Variety and quality of in-flight meals and drinks | A | O | I | A |
| SQA12. Variety and quality of in-flight entertainment | I | I | I | I |
| SQA13. Access to Wi-Fi | I | I | I | A |
| SQA14. Access to in-seat power | A | A | I | A |

OTP is a one-dimensional attribute for all Clusters. The finding confirms that punctuality is an important aspect for business travellers because they are time-sensitive and flight delays can mean missed appointments [80].

FFPs are considered a must-be SQA for Clusters one, two, and four, while Cluster three regards the FFP as an attractive attribute. This indicates that business travellers flying on short-haul in economy class do not expect the provision of a FFP, but will be satisfied when it is provided. As suggested by Huse and Evangelho [81] and Mikulić

and Prebežac [46], flying with LCCs makes business travellers re-evaluate the importance placed on certain SQAs.

The cleanliness of the aircraft interior and seats were classified as a must-be SQA by Clusters one, two, and three, and as one-dimensional by Cluster four. This result conforms to Tahanisaz and Shokuhyar [19], revealing that travellers will be dissatisfied when airlines provide insufficient cabin cleanliness.

Business travellers in Clusters one, three, and four are indifferent about the appearance of the cabin crew. The research by Chen and Chang [13] revealed that passengers considered cabin crew appearance less important and placed more importance on the cabin crew's level of professionalism and service proficiency. The attribute politeness of cabin crew is considered one-dimensional for Clusters one, two, and four, increasing business traveller satisfaction. This finding conforms with Liou et al. [5] who found that flight attendant courtesy and willingness to help are rated as the most important determinant of service quality.

The in-flight entertainment SQA is regarded as an indifferent attribute by all Clusters. Therefore, not contributing towards business traveller satisfaction. This finding is in line with the previous literature and conflicts with the International Airport Review study in 2018 which revealed that in-flight Wi-Fi influences the loyalty and satisfaction of business travellers and that 90% of business travellers indicated their intention to use the Wi-Fi service on their next flight.

Business travellers who fly economy class on short-haul do not consider the provision of in-seat power as important. Nevertheless, in-seat power is considered an attractive SQA for Clusters one, two, and four. Although this contradicts the findings of Tahanisaz and Shokuhyar [19], revealing that business travellers identified in-flight electricity as a must-be attribute, it is consistent with the findings of Shahin and Zairi [66].

*4.3. Importance Satisfaction Analysis*

To demonstrate the importance and the degree of satisfaction of the SQAs simultaneously, the results were plotted on the ISA matrix for each Cluster. An SQA is satisfactory if the satisfaction rating is positive and unsatisfactory if the rating is negative. Figure 8 illustrates the importance satisfaction matrix of the four Clusters concerning the fourteen selected SQAs. All the attributes scored a satisfactory/positive rating for all Clusters except for SQA4 for Cluster two. Hence, travellers who fly business class on long-haul flights consider airline performance to be satisfactory, except for airline responsiveness to complaints.

In the 'concentrate here' quadrant, SQA6 and SQA7—both on the tangible dimension—are common across all Clusters. Business travellers consider the attributes of seat comfort and the cleanliness of aircraft interiors and seats as highly important. However, the performance of airlines in these attributes is regarded as below average, leading to low satisfactory levels implying that efforts to improve these areas should be concentrated on. In the 'keep up the good work' quadrant, SQA2 (empathy dimension) and SQA3 (assurance dimension) are common across the four Clusters. Business travellers, regardless of the haul of trip and cabin category, consider the attributes of OTP and FFP as highly important, with high satisfaction levels. Consequently, airlines should maintain their performance in these attributes.

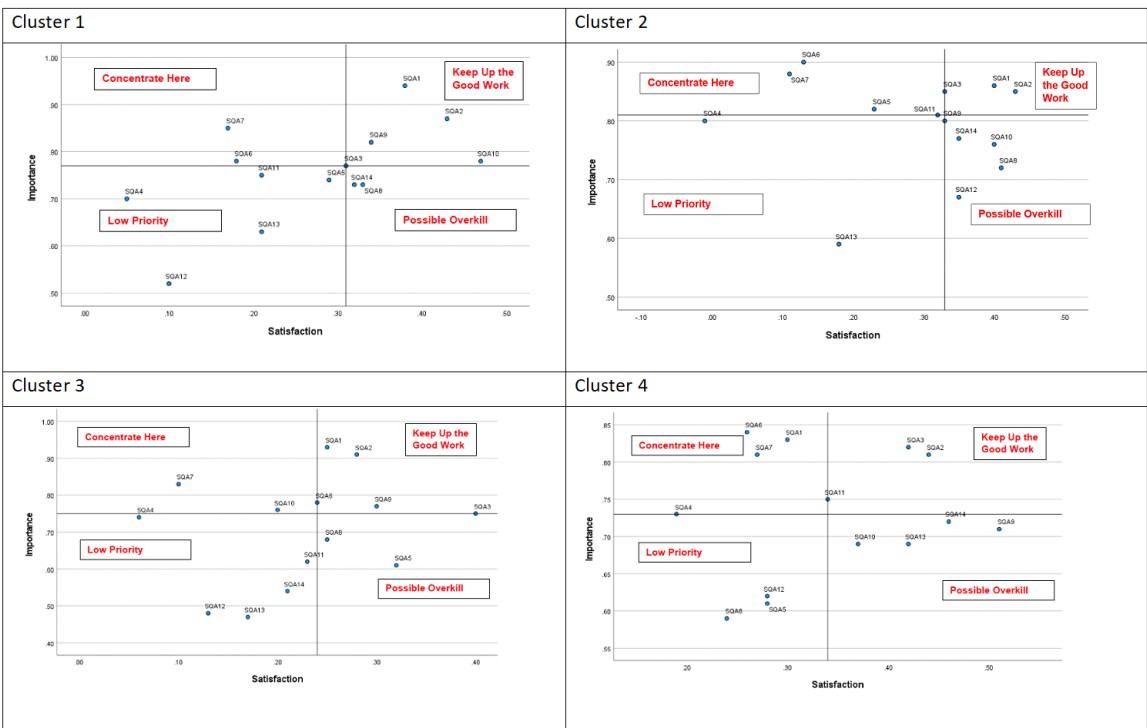

**Figure 8.** Importance satisfaction analysis for four business traveller Clusters.

SQA4 (reliability dimension) is located in the 'low priority' quadrant. However, responsiveness to complaints for three Clusters is located very near the threshold of the 'concentrate here' section. This attribute is only regarded as weakly satisfactory and dissatisfactory for Cluster two. Additionally, SQA12 in-flight entertainment (tangible dimension) is situated in the 'low priority' quadrant for three out of four Clusters.

The final quadrant 'possible overkill' includes SQA14 in-seat power (tangible dimension) as a common attribute for all the business traveller Clusters included in the study. This signifies that resources should be reallocated somewhere else.

### 4.4. The Airline Manager's Perspective

To determine whether airline managers are resonating with the requirements of business travellers, four airline managers from British Airways, easyJet, Edelweiss Air, and Etihad Airways were provided with the same questionnaire to give their responses concerning the fourteen SQAs in terms of five categories of the Kano model and their ISA rating.

Airline managers resonated with business travellers' classification as 'important' regarding the following attributes: convenient flight schedules, cabin cleanliness, politeness of cabin crew, in-flight refreshments, and entertainment. However, they classified FFP as an attractive attribute, unlike travellers in Clusters 1, 2, and 4. Additionally, airline managers categorised the lounge service as indifferent for economy class passengers and as important for business class travellers.

Regarding the importance–satisfaction analysis (ISA), airline managers concur with the sampled business travellers regarding responsiveness to complaints, politeness of cabin crew, knowledge and experience of cabin crew, and in-flight entertainment. Additionally, they consider that convenient flight schedules and OTP are highly important for business travel, which is in line with all Cluster responses (Figure 9).

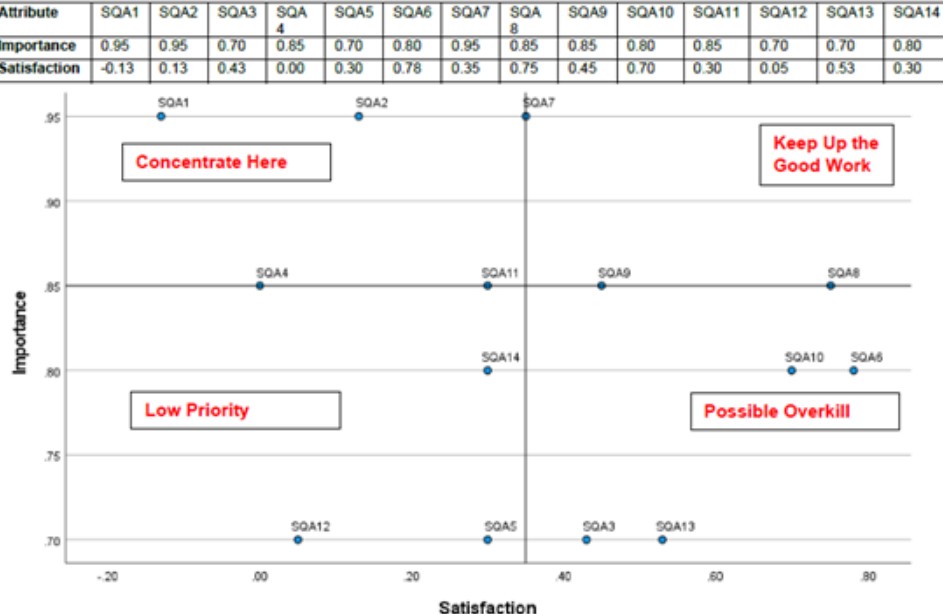

| Attribute | SQA1 | SQA2 | SQA3 | SQA4 | SQA5 | SQA6 | SQA7 | SQA8 | SQA9 | SQA10 | SQA11 | SQA12 | SQA13 | SQA14 |
|---|---|---|---|---|---|---|---|---|---|---|---|---|---|---|
| Importance | 0.95 | 0.95 | 0.70 | 0.85 | 0.70 | 0.80 | 0.95 | 0.85 | 0.85 | 0.80 | 0.85 | 0.70 | 0.70 | 0.80 |
| Satisfaction | -0.13 | 0.13 | 0.43 | 0.00 | 0.30 | 0.78 | 0.35 | 0.75 | 0.45 | 0.70 | 0.30 | 0.05 | 0.53 | 0.30 |

**Figure 9.** Airline manager importance–satisfaction analysis.

Airline managers allocated seat comfort in the 'possible overkill' quadrant, unlike passengers in all Clusters that considered this attribute as important. Notably, one of the managers emphasised that business travellers often use long-haul flights to rest overnight and go straight to work the next day. Additionally, airline managers allocated cabin cleanliness into the 'keep up the good work' quadrant; however, business travellers in all Clusters assigned this attribute into the 'concentrate here' quadrant.

## 5. Conclusions and Managerial Implications

This study identified fourteen SQAs as perceived by business travellers in four Clusters. The attributes are distributed across four dimensions of airline service quality i.e., "reliability, assurance, tangibles, empathy and responsiveness", and captured relative importance and satisfaction levels using the Kano model and importance–satisfaction analysis approach.

The study disclosed the efficient and effective resource allocation for only five SQAs, namely convenient flight schedules and frequencies, OTP, FFP, politeness of cabin crew, and knowledge and experience of cabin crew, whereas five other SQAs, namely airline lounge services, the appearance of cabin crew, variety and quality of in-flight entertainment, in-flight Wi-Fi, and in-seat power availability, required management attention based on passenger evaluations suggesting that resource allocations were either inadequate or in the overkill category. The findings regarding airline managers' opinions on SQAs revealed that they generally resonate with business travellers regarding several SQAs. However, differences between the assessment of attributes by business travellers and airline managers were acknowledged. This included FFPs, and in-flight Wi-Fi, which were classified as attractive attributes by airline managers. The findings indicate that this is in line with Cluster four; however, all other Clusters were shown to be indifferent about in-flight Wi-Fi. Additionally, airline managers allocated seat comfort in the 'possible overkill' quadrant, whereas the majority of Clusters placed that attribute in the 'concentrate here' quadrant.

As a result, it is clear that managers need to thoroughly examine these criteria and allocate resources to service quality dimensions with higher priority, based on the most relevant Cluster segments flying the specific route.

This study offers a framework of necessary steps to capture service expectations of airline business passengers and identifies the areas where service quality needs to be improved, and accordingly, where resources need to be allocated based on customer

expectations. This is expected to have a positive impact on airline relational capital, customer loyalty, and, consequently, on profitability, given that business travellers are considered twice as profitable as leisure travellers.

When taking the results of the Clustering, Kano Model, and ISA into consideration, the following specific managerial implications can be made to optimise resource allocation and improve service quality and business traveller satisfaction.

1. Maintain performance regarding the SQAs of convenient flight schedules and frequencies, OTP, FFPs, and politeness of cabin crew;
2. Concentrate on improving the responsiveness to complaints and the cleanliness of the aircraft interior and seating. Enhance lounge services for business class travellers;
3. Focus on improving seat comfort in business class as well as in the long-haul premium economy and economy class. Contrarily, keep up the seat comfort in the short-haul economy class;
4. Concentrate on enhancing in-flight food and beverages in business class. Maintain performance in economy class long-haul and replace a short-haul meal service focus in economy class with a buy-on-board service;
5. Allocate more limited resources to in-flight entertainment and in-flight Wi-Fi. FSNCs should maintain the provision of reliable power outlets and USB ports to positively attract business travellers. Contrarily, LCCs should not overkill on highlighting the benefits of in-seat power and should allocate resources elsewhere.

Thus, the results would assist airline management to prioritise areas for improvement and allocate resources more effectively. Additionally, the managerial implications would be in using the findings to inform marketing strategies and messaging, highlighting the areas of service quality that are most important to business travellers and differentiating the airline from competitors.

This study has some limitations that present opportunities for further research. One limitation is the reliance on a convenience sample of business travellers. To add further validity and improve the insights into the perceptions of business travellers, future studies could use contrasting research methods and/or sampling methods, such as personal interviews.

The analysis shows that satisfaction and specific service quality attributes (SQAs) are important for airlines, but improvement needs and priorities may vary between airlines. Future research should gather data specifically from business travellers for each airline for a more comprehensive understanding. Additionally, future research should include more SQAs for wider coverage and should also gather data from a larger sample of airline managers for a more representative understanding of their perceptions.

**Author Contributions:** Conceptualisation, P.L. and J.F.O.; methodology, P.L. and J.F.O.; formal analysis, P.L., N.I., J.F.O., D.W.S. and M.E.; data curation, P.L., J.F.O. and N.I.; writing—original draft preparation, P.L.; writing—review and editing, N.I., J.F.O., D.W.S. and M.E. All authors have read and agreed to the published version of the manuscript.

**Funding:** This research received no external funding.

**Data Availability Statement:** Data is available upon request by writing to the corresponding author.

**Conflicts of Interest:** The authors declare no conflict of interest.

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
