# Peer review of "Investigating Airline Service Quality from a Business Traveller Perspective through the Integration of the Kano Model and Importance–Satisfaction Analysis"

_sustainability, doi:10.3390/su15086578_

Round 1

Reviewer 1 Report

Dear Author(s), 

Thank you for your paper. I would like to underline that I found the topic interesting. These are some revisions suggested, as I believe this paper would be a great fit for the journal: 

The paper need more careful editing, for example, uniformed the way the same term is written ( 111 SERVQUAL and 161 Servqual)

What was the timing of the survey?

Was it the EU population?

What was the software used for data analysis?

240 What was the logic of “The targeted sample size is 250 respondents”?

What is the sampling method?

Did the authors proceed with the pre-test? What was the result?

Did the authors show the results of the quantitative survey to the managers? What was the opinion?

It is highly desirable to have more updated references.

Needs a better link to the journal subject.

Thank you again for your work and I wish you all the best in your future research!

Author Response

Thank you for the constructive comments. Replies are enclosed in the attached document.

Best regards.

Reviewer 2 Report

This paper surveyed business travellers about airline service quality using the Kano Model combined with Importance Satisfaction Analysis. The paper analyzed the airline service quality systematically despite its complex nature, with appropriate standards and methods. However, the following three improvements are recommended:

1. The existing well-organized charts need to be supplemented with a table organized by service quality attributes that shows the whole picture once.

2. Poor readability due to editing style such as alignment (e.g., SQA4, SQA8 in the table in Figure 9, etc.) needs to be fixed by reviewing and making changes.

3. Letters in figures that are illegible in printouts need to be checked and edited.

Author Response

(The authors gave the same response as above.)

Reviewer 3 Report

Dear Authors

I have read the article"Investigating airline service quality from a business traveller perspective through the integration of the Kano model and Importance-Satisfaction analysis". I liked the article, however, I have a few suggestions to make it better.

1. Either US or UK English to be followed. I found both: e.g., behavior and behaviour. CHECK.

2. Only 1 citation from 2022 and 3 from 2021. Recency in citation must be there.

3. Introduction: Background is not strong enough. Logical flow is needed. Section plan is missing.

4. Before literature review, theoretical background with base theory must be provided.

5. First, Define all the constructs before proposing the relationships.

6. "Data is collected using both primary and secondary data sources." How did you use the secondary data???

7. Instrument design is done before data collection. It is common sense. But, you explain later....CHANGE.

8. You say "The target segment is business travellers worldwide, with no geographic limitations. The targeted sample size is 250 respondents, while the number of responses collected from the questionnaire is 339 responses with 265 being acknowledged as usable and utilised. Additionally, usable responses from four airline managers are collected."--

What is your sample size????

There is  no defined population. Anyone??? no geographic limitations??? millions??? and 250 samples??? VERY BAD SAMPLE DESIGN

9. Questionnaire used in the study has to be checked, not the items. Attach as an appendix.

10. For SEM based studies, cite latest studies:

Dash, G., & Paul, J. (2021). CB-SEM vs PLS-SEM methods for research in social sciences and technology forecasting. Technological Forecasting and Social Change173, 121092.

11. The study is a good one, hence it must have theory and policy implications too.

All the best.

Author Response

Thank you for the constructive comments. Below are the responses to the comments raised. Edits has been reflected in the updated manuscript.

  • US and UK English corrected. Now all is UK English.
  • More recent references are added, for example – Sangwon Park, Jin-Soo Lee, and Juan L. Nicolaub (2020).
  • Section plan added towards the end of the introduction.
  • Secondary data sources are used to produce the literature review and to identify the research gap. Additionally, it was used to validate the findings of the primary research through comparing the results of the survey with the findings from previous research (secondary sources).
  • The authors aimed to accomplish a minimum sample size of 250. 339 responses were obtained from the questionnaire, 265 were acknowledged as usable and were therefore utilised. Hence, the sample size of 265 was considered adequate and valid.
  • This study uses convenience sampling, that is taken from a segment of the population that is accessible. Even though the data collected from a convenience sample does not allow definite findings and the research cannot be generalised (Bryman and Bell, 2015), convenience sampling utilising the internet, enables the researchers to reach the difficult to locate population of Business Travellers.
  • SEM latest studies cited - Dash, G., & Paul, J. (2021). CB-SEM vs PLS-SEM methods for research in social sciences and technology forecasting. Technological Forecasting and Social Change, 173, 121092.

Reviewer 4 Report

Reviewer Report

Thanks for giving me a chance to review this manuscript. This is an interesting topic. The author(s) tries to work significantly. The author (s) aims to use the Kano Model and Importance Satisfaction Analysis (ISA) to assess airline service quality by identifying the prioritized Service Quality Attributes (SQA) for business travelers. However, still, some of the anomalies I found during the review process are addressed below, which may help further develop the study.

Originality

It can be improved

The author(s) should illustrate how this research differs from the literature in the field / or considered a contribution to the literature

Relationship to Literature

It can be improved

-           More studies relevant to the field should be cited and used in this paper, review papers should be helpful, for example:

•           Ashutosh Pandey, Rajendra Sahu & Yatish Joshi (2022) Kano Model Application in the Tourism Industry: A Systematic Literature Review, Journal of Quality Assurance in Hospitality & Tourism, 23:1, 1-31, DOI: 10.1080/1528008X.2020.1839995-     The previous gaps in the literature are not evident, they should be illustrated

-           While the author(s) focused on the theoretical gap (gap in the literature), they should explain the practical gap that could help to justify conducting this study.  

-           “Nonetheless, there are relatively few studies in the airline industry that assess the comparative significance of service quality aspects on the satisfaction levels of business travelers” , … “There are very few studies investigating the importance of airline SQAs exclusively from a business traveler perspective.”

Here it is recommended to cite some of those studies.

Methodology:

It can be improved

-           In the 3rd section (3. Methodology), the author should conclude the section by addressing why this methodology is suitable to the research context.

-           The study population and sample should be well explained in this section, and why “passengers who previously traveled on commercial aeroplanes for work or business purposes” is considered a good sample for this study. Also, the author(s) need to justify the number of responses that are considered sufficient to be used in this study  

Results and discussion

It can be improved

-           In the discussion section, the author(s), need to discuss the findings compared with previous research on the field and cite other scholars' work

Quality of Communication:

-           There is a need for proofreading and professional editing.

Additional Comment

·       Theoretical implications should be added

·       The “limitations and future work” section should be added 

Author Response

Thank you for the constructive comments. Below are the responses to the comments raised. Edits has been reflected in the updated manuscript.

  • How the research differs from the literature is being highlighted - There are very few studies investigating the importance of airline service quality exclusively from a business traveller perspective. This study adds to the limited existing research providing airline managers with information on business passengers’ expectations, their level of fulfilment, and the impact of these expectations on airline operator resource capacity and allocation.
  • Panday et. al., 2022 study has been added to the citations for literature review.
  • Two studies investigating service quality from business traveller perspective are cited Ref. No. 17 and No. 18.
  • Instrument design and sampling technique sections have been improved.
  • Study limitations and further research edited in section five.

Reviewer 5 Report

Dear Authors,

The submitted manuscript is an interesting study on investigating airline service quality from a business traveller perspective. The research procedure was correctly applied to obtain, process and interpret the results. The results have not only a practical dimension, but also a substantive dimension, extending the knowledge of social phenomena and a methodological dimension in terms of research methodology.

After studying the text, the following comments emerge:

In Table 1 the attributes should accordingly be better organised considering the Service Quality Dimensions groups. However, as this would result in the need to organise the data in subsequent stages of the research investigation, please take this comment into consideration when compiling other data in the future.

In line 254 - the reference to Table 1 should precede Table 1.

In line 262 - are you sure this is a Likert scale or rather a semantic scale? This is a common factual error repeated in many publications.

In line 310 - at the end of the sentence "...of respondents out of the four clusters" it is essential to refer to (Figure 7).

In line 318 - in the sentence after "...that they commonly travel with a LCC" it is imperative to refer to (Figure 7) as the interpretation continues.

Author Response

(The authors gave the same response as above.)

Round 2

Reviewer 3 Report

Thanks for accepting the suggestions and the defense.

All the best

Author Response

Thank you for your review and for being critical. It helped in improving the rigour of the study.